# Comparison of Humoral Response between Third and Fourth Doses of COVID-19 Vaccine in Hemodialysis Patients

**DOI:** 10.3390/vaccines11101584

**Published:** 2023-10-12

**Authors:** Yoosun Joo, Dae Kyu Kim, Yun Gi Jeon, Ah-Ra Kim, Hyeon Nam Do, Soo-Young Yoon, Jin Sug Kim, Su Woong Jung, Hyeon Seok Hwang, Ju-Young Moon, Kyung Hwang Jeong, Sang-Ho Lee, So-Young Kang, Yang Gyun Kim

**Affiliations:** 1Division of Nephrology, Department of Internal Medicine, Kyung Hee University College of Medicine, Kyung Hee University Hospital at Gangdong, Seoul 05278, Republic of Korea; yunisysjoo@gmail.com (Y.J.); ha-ppy@hanmail.net (S.W.J.); jymoon@khu.ac.kr (J.-Y.M.); lshkidney@khu.ac.kr (S.-H.L.); 2Division of Nephrology, Department of Internal Medicine, Kyung Hee University College of Medicine, Kyung Hee University Medical Center, Seoul 02447, Republic of Korea; kevinkim222@naver.com (D.K.K.); lynnyoon41@gmail.com (S.-Y.Y.); jinsuk0902@naver.com (J.S.K.); hwanghsne@gmail.com (H.S.H.); aprilhwan@naver.com (K.H.J.); 3Honorshill Hospital, Gimpo-si 10035, Republic of Korea; cyk0404@hanmail.net; 4Division of Vaccine Clinical Research Center for Vaccine Research, National Institute of Infectious Diseases, Cheongju 34142, Republic of Korea; kara9251@korea.kr (A.-R.K.); hndo@korea.kr (H.N.D.); 5Department of Laboratory Medicine, Kyung Hee University College of Medicine, Kyung Hee University Hospital at Gangdong, Seoul 05278, Republic of Korea; sykangmd@daum.net

**Keywords:** COVID-19, hemodialysis, humoral immunity, vaccination

## Abstract

Dialysis patients are more likely to die or become hospitalized from coronavirus disease 2019 (COVID-19). Currently, only a few studies have evaluated the efficacy of a fourth booster vaccination in hemodialysis (HD) patients and there is not enough evidence to recommend for or against a fourth booster vaccination. This study compared the humoral response and disease severity of patients on HD who received either three or four doses of COVID-19 vaccine. A total of 88 patients were enrolled. Humoral response to vaccination was measured by quantifying immunoglobulin G levels against the receptor binding domain of SARS-CoV-2 (anti-RBD IgG) at five different times and plaque reduction neutralization tests (PRNT) at two different times after vaccination over a period of 18 months. Antibody levels were measured at approximately two-month intervals after the first and second dose, then four months after the third dose, and then one to six months after the fourth dose of vaccine. PRNT was performed two months after the second and four months after the third dose of vaccine. We classified patients into four groups according to the number of vaccine doses and presence of COVID-19 infection. Severe infection was defined as hospital admission for greater than or equal to two weeks or death. There was no difference in antibody levels between naïve and infected patients except after a fourth vaccination, which was effective for increasing antibodies in infection-naïve patients. Age, sex, body mass index (BMI), dialysis vintage, and presence of diabetes mellitus (DM) did not show a significant correlation with antibody levels. Four patients who experienced severe COVID-19 disease tended to have lower antibody levels prior to infection. A fourth dose of SARS-CoV-2 vaccine significantly elevated antibodies in infection-naïve HD patients and may be beneficial for HD patients who have not been previously infected with SARS-CoV-2 for protection against severe infection.

## 1. Introduction

Patients with end-stage renal disease (ESRD) who are undergoing hemodialysis (HD) are more susceptible to death from coronavirus disease 2019 (COVID-19) compared to the general population. Multiple factors contribute to the increased susceptibility of dialysis patients. First, patients with renal failure have an impaired immune response due to decreased phagocytic function of polymorphonuclear cells, defective antigen presentation and costimulatory activity, and reduction in dendritic cells and B cells [1,2,3,4]. These patients also have increased systemic inflammation due to upregulation in Toll-like receptors and increased production of pro-inflammatory cytokines, as well as oxidative stress via increased reactive oxygen species production [5,6]. Second, HD patients tend to be older and have comorbidities contributing to immune impairment, such as diabetes [7]. Third, current knowledge suggests that patients with ESRD show impaired cellular and humoral immune response to severe acute respiratory syndrome coronavirus 2 (SARS-CoV-2) vaccination [8,9]. Real-life vaccine effectiveness is also lower in dialysis patients [10]. Lastly, chronic HD patients are at increased risk of exposure since they must travel to dialysis centers and contact other patients and healthcare professionals during routine dialysis visits [7].

During the COVID-19 pandemic, because ESRD patients were considered to be a high-risk population, there was a need to prioritize vaccination. One solution for this issue was heterologous vaccination. In retrospect, patients with ESRD who received heterologous vaccination showed antibody levels that were non-inferior to those of the general population [11]. Hence, heterologous vaccination may be a tool for achieving stronger and faster immunity against COVID-19, especially in countries with vaccine shortages [12,13]. Monitoring the response to vaccination is important in the dialysis population because patients with ESRD tend to have weaker antibody responses to vaccination [14]. The level of antibodies against SARS-CoV-2 decreases over time [15,16], and low antibody levels at approximately six months following vaccination have been shown to increase the risk of breakthrough infections [17]. Lower antibody levels are associated with higher hazard ratios for infection, COVID-19 related hospitalization, or death [18,19].

Previous studies have already demonstrated the utility and cost-effectiveness of a third SARS-CoV-2 vaccine for increasing anti-SARS-CoV-2 spike IgG antibodies and neutralizing antibodies [20,21,22]. However, there is currently insufficient evidence to validate the need for a fourth or fifth booster dose. Emerging evidence shows that a fourth booster vaccine increases antibody response and reduces mortality in HD patients, but only few studies consider differences in vaccine response between COVID-19-infected and naïve dialysis patients as well as the dynamics of antibody response throughout the vaccination course [23,24,25]. Therefore, we examined the effect of a fourth SARS-CoV-2 vaccine for boosting antibody response in HD patients. We also evaluated the protective effect of a fourth SARS-CoV-2 vaccine in reducing disease severity in ESRD patients by (1) serial measurement of antibody levels and neutralizing antibodies and (2) comparison of patients who received either three or four doses of vaccine, along with COVID-19 infection status.

## 2. Materials and Methods

A total of 88 patients who received maintenance HD twice or thrice a week at Kyung Hee Medical Center and Kyung Hee University Hospital at Gangdong were enrolled. Enrolled patients were between ages 18 to 90 and received maintenance dialysis for greater than three months. Patients with a life expectancy of less than six months, active malignancy, and those who refused vaccination were excluded from the study. Patients were categorized into four groups based on the total vaccine dose received and whether they were infected with COVID-19. Twenty-eight patients who were COVID-19-naïve and 23 patients who were infected with COVID-19 received three doses of the vaccine while another 28 patients who were COVID-19-naïve and nine patients who were COVID-19-infected received four doses of the vaccine (Figure 1). A positive real-time polymerase chain reaction (PCR) test for COVID-19 at any time during the study period was used to classify patients as COVID-19-infected, where a positive test result was defined as a Ct value of less than 36 for the envelope (E) gene or the RNA-dependent RNA polymerase (RdRp) gene of SARS-CoV-2. Patients who did not have a positive PCR result during the entire study period were classified as COVID-19-naïve. All patients received heterologous vaccination with ChAdOX1 and BNT162b2 at two-month intervals. The third dose of the vaccine was identical in all patients (BNT162b2) and was administered four months after the second dose. Lastly, five to six months after the third dose of the vaccine, some patients received a fourth vaccine with either BNT162b2, CX-024414, or NVX-CoV2373.

Baseline clinical information and demographic data were collected within one month before the first vaccination dose through a retrospective review of electronic medical records. For the primary outcome, serum immunoglobulin G against the receptor-binding domain of the S1 subunit of the spike protein of SARS-CoV-2 (anti-RBD IgG) was measured at five different time points, and neutralizing antibodies against SARS-CoV-2 were measured at two different time points. Correlation analyses were performed for the following factors: age, BMI, sex, dialysis vintage, and presence of DM. The anti-RBD IgG assay was quantified using the ARCHITECT IgG II Quant test (Abbott Laboratories), which is an automated two-step chemiluminescent microparticle immunoassay. Antibody levels greater than or equal to 50.0 AU/mL were defined as a positive result. Quantification of anti-RBD IgG was performed 2 months after the first dose (T1), 2 months (T2) and 4 months after the second dose (T3), 4 months after the third dose (T4), and 1–6 month after the fourth dose of vaccine, or 12 months after the third dose (T5) for all groups, as outlined in Table 1. Neutralizing antibodies against SARS-CoV-2 were measured using the plaque reduction neutralization test (PRNT). Neutralizing antibodies were quantified at T2 and T4 in 15 randomly selected COVID-19-naïve patients. All test sera were heat-inactivated at 56 °C for 30 min, then serially diluted. The diluted sera were incubated with 50 plaque-forming unit of SARS-CoV-2 virus (hCoV-19/Korea229079/KDCA/2021, hCoV-19/Korea/KDCA447321/2021) for 1 h at 37 °C and 5% CO_2_. The virus–serum mixtures were added to confluent of Vero cell in a 12-well plate and incubated at 37 °C, 5% CO_2_ for 1 h with shaking every 10 min. The virus–serum mixtures were removed and 1 mL of overlay medium (0.75% agarose, 2% 2X MEM) was added. The plate was incubated at 37 °C, 5% CO_2_ for 3 days. At the end of the incubation period, cells were fixed and stained with a crystal violet mixture (0.07% crystal violet, 10% formaldehyde solution, 5% ethanol) for 5–6 h at room temperature, the overlaid agar was decanted, and the plaques were visualized. The 50% neutralizing dose (ND_50_) was calculated using the Karber formula. For the secondary outcome, the severity of COVID-19 was assessed, where severe disease was defined as hospital admission for greater than or equal to two weeks or death. The public health policy in Korea during the COVID-19 pandemic was to transfer all confirmed HD patients to designated dialysis centers for one week. If the disease course was uneventful and patients showed no complications, patients returned to their original dialysis center after one week. Therefore, we defined severe COVID-19 as patients who were hospitalized for more than two weeks to reflect complicated cases that required prolonged treatment.

This study was conducted in accordance with the Declaration of Helsinki, and the participants provided written informed consent prior to the start of the study. This study was reviewed and approved by the Clinical Institutional Review Board of Gangdong Kyung Hee University Hospital (IRB number: 2021-11-013-004).

Data were expressed as means ± standard deviation (SD) or median (interquartile range) and were compared using Mann–Whitney test or the Wilcoxon rank-sum test. Differences among the three groups were identified using analysis of variance or the Kruskal–Wallis test. Chi-square test or Fisher’s exact test was used to compare categorical data. Spearman’s correlation test was used to evaluate the correlation between anti-RBD IgG and continuous variables. The association between anti-RBD IgG and categorical variables was evaluated using linear regression analysis. The ID50 of the PRNT and anti-RBD IgG titers were compared using Spearman’s correlation test. All *p* values were two-tailed, and statistical significance was set at *p* < 0.05. Statistical analyses were performed using the SPSS (version 22.0; IBM Corp., Armonk, NY, USA) and R (version 4.0.0) software.

## 3. Results

### 3.1. Baseline Characteristics and COVID-19 Infection

Among the patients who received three doses (*n* = 51), 23 were later infected with COVID-19 and 28 were not infected during the study period. Among the patients who received four doses (*n* = 37), nine were diagnosed with COVID-19. Baseline characteristics were similar among four groups, except for age, where COVID-19-naïve patients who received four doses of the vaccine were significantly older compared to COVID-19-naïve patients who received three doses (*p* < 0.05) (Table 2). Other characteristics such as serum albumin level, dialysis adequacy (Kt/V), and dialysis vintage were not significantly different among the four groups.

### 3.2. Anti-RBD IgG Levels and Neutralizing Tests

In COVID-19-naïve patients who received three doses of vaccination, anti-RBD IgG levels sequentially increased after the third vaccination and then decreased at the final measurement at 12 months after the third vaccination; for COVID-19-naïve patients who received four doses, antibody levels continued to increase until after the last vaccine dose (Figure 2a). While there was no difference in the antibody levels of COVID-19-infected patients (regardless of vaccine dosage), COVID-19-naive patients who received four doses of vaccination had significantly higher antibody levels compared to naive patients who only received three doses (*p* < 0.05) (Figure 2b and Table 3). Correlation analyses were performed for age, BMI, dialysis vintage, sex, and presence of diabetes mellitus. Overall, there was no correlation between antibody levels and subgroups for all time points, except for male sex at T1 (Table 4). PRNT was performed to quantify neutralizing antibodies against Delta and Omicron variants in 15 randomly selected infection-naïve patients. Anti-RBD IgG levels correlated with neutralizing antibody levels in all cases except for the Omicron variant at T2 (Figure 3).

### 3.3. Anti-RBD IgG Levels and Disease Severity

Four patients were classified as severe, with two fatal cases and two cases requiring hospitalization for greater than two weeks, as summarized in Table 5. All patients were infected after their third dose of vaccine. The two deceased patients were diagnosed with COVID-19 in March 2022, and deaths occurred on 22 March and 28 March 2022. In these patients, anti-RBD IgG levels measured directly prior to infection were 342 and 317 AU/mL, whereas the mean value in non-severe COVID-19 patients was 2870.7 AU/mL and 1905.6 AU/mL for non-infected patients. Anti-RBD IgG levels in severe cases (which required hospitalization for longer than two weeks) were 553.2 and 788.8 AU/mL. Comparing antibody levels in severe and non-severe patients showed a statistically significant difference at both T2 and T3, whereas severe and non-infected patients only showed a difference at T3 (Figure 4). Deceased patients were censored after T3, and statistical analysis was not possible because the severe group only consisted of two patients.

## 4. Discussion

The average age of the COVID-19-naïve patients who received four doses of vaccination was much older than other groups. Although age is not a determining factor for vaccination, this pattern is likely associated with multiple social factors. First, patients older than 60 years were considered a high-risk group and were reminded more frequently to receive booster COVID-19 vaccinations via short message services at the population level. Second, as the spread of COVID-19 is correlated with community exposure, younger, socially active patients were exposed to COVID-19 during the first wave of Omicron infection (February 2022 to May 2022), while older, less active patients remained uninfected (Figure 4) [26]. Patients who were not infected with COVID-19 were more likely to receive the fourth dose of COVID-19 vaccination (40.0% vs. 29.4%). Third, patients were not as wiling to receive a fourth vaccine. Patient reluctance toward the fourth vaccine was multi-faceted, owing to vaccine fatigue, suspicion about vaccine reliability, and mass media coverage reporting Omicron variant as being less deadly than previous variants. As a result, the acceptance and timing of a fourth vaccine was strictly voluntary. Factors such as age, BMI, sex, and dialysis vintage did not show a consistent correlation with anti-RBD IgG levels. Previous studies showed age and presence of diabetes mellitus (DM) as factors associated with response to vaccination, but analysis of our data did not show a significant relationship between antibody formation and DM [27]. The reason for lack of correlation between age and antibody formation can be attributed to the small number of sample sizes that included patients who recovered from COVID-19, making the effects of natural immunity a potential confounding factor. However, comparison of COVID-19-infected and -naïve patients did not show a consistent relationship between age and antibody levels, and larger studies with a greater sample size are needed to validate this relationship. Contrary to the results of previous studies, other factors associated with the response to vaccination, such as albumin, Kt/V, and dialysis vintage, were not significantly correlated with antibody levels all time points in our study [14,28].

Four patients with severe COVID-19 (two hospitalizations and two deaths) tended to have lower antibody levels than those with non-severe infections. In particular, the two fatal cases showed markedly low levels of anti-RBD IgG directly prior to infection. Korean epidemiologic data show that there were two waves of COVID-19 infection. The first wave occurred from February 2022 to May 2022, which corresponds to the time interval between T3 and T4, and the second wave occurred from July 2022 to October 2022, which corresponds to the time between T4 and T5 (Figure 5). In our study, all severe and fatal cases occurred between T3 and T4 stages. None of the patients were on immunosuppressive medications, and patient characteristics were not different between severe, non-severe, and non-infected cases in terms of age. Although the number of cases is too small to make a statistically significant conclusion, we observed that dialysis patients with lower antibody levels are more likely to experience severe disease after COVID-19 infection. This observation concurs with the results of other studies showing that lower antibody levels are associated with hospitalization and mortality [29].

Examination of the anti-RBD IgG trend in all patients reveal that antibody levels show a positive trend until the third vaccination and continue to rise in patients who receive four doses, whereas patients who only receive three doses show a decreasing trend. The same trend in antibody levels was observed in other studies comparing the third and fourth doses of SARS-CoV-2 vaccination in patients with ESRD [30]. COVID-19-naïve patients who received four doses had significantly higher antibody levels than those who received three doses only. In the general population, vaccine effectiveness starts to decrease at two to four months following vaccination [31]. Antibody levels slowly fall over a period of six months after natural infection and decrease substantially at six months following vaccination in HD patients [32,33]. Sequential data show that antibody levels decrease at two months after the second vaccine (between T2 and T3) and seven months (between T4 and T5) after the third vaccine (Figure 2a). Although the reduction in antibody levels is dependent on the number of vaccinations, it can be roughly estimated that antibodies decrease at about three to six months following vaccination. Based on the observation of severe and fatal patients in this study, as well as other studies evaluating antibody level and disease course, it is possible to infer that the humoral response achieved through additional vaccination confers protection for severe disease and mortality [34]. In patients who previously had low response to vaccination, the third or fourth dose significantly increased the immunogenicity rates [35]. Similarly, patients who were weak responders to the second dose of the vaccine showed a greater increase in antibody levels after the third dose, suggesting that receiving a fourth booster vaccination is advantageous for patients with ESRD [36]. When comparing anti-RBD IgG levels between the infected and non-infected groups, asymptomatic COVID-19 may be a confounding factor that increases anti-RBD IgG levels. However, there was nationwide surveillance for fever and symptoms of upper respiratory tract infection, and COVID-19 PCR was performed in patients with a low index of suspicion, making the inclusion of asymptomatic infection unlikely.

In COVID-19-naïve patients, anti-RBD IgG levels were interpreted as a marker of immune response to vaccination, whereas in COVID-19-infected patients, it was interpreted as a marker of natural immunity. Although serum immunoglobulin levels are a correlate of protection for the original virus strain from Wuhan, China, their role as a correlate of protection for variants of concern (VOCs) such as Delta and Omicron, remains inconclusive [37,38,39]. To determine the protective effect of anti-RBD IgG against VOCs, the ID_50_ of neutralizing antibodies was compared with anti-RBD IgG levels, and the association was significant for all variants except for the Omicron variant at two months after the second dose (T2). This result indicates that humoral immunity of booster vaccination offers some degree of protection against variants. Because vaccine-induced and infection-induced memory generate antibodies that differ in neutralization capacity, only the sera from COVID-19-naïve patients were used for the analysis [40,41]. This sample was frozen until it could be pooled for analysis, which may explain why Figure 3a did not show a significant correlation. Another potential cause of the low level of correlation may be due to less robust anti-RBD IgG levels following second vaccination in Figure 3c.

Currently, there is no established definition of an antibody level that confers seroprotection [42]. Affeldt et al. showed a cutoff value of 296 AU/mL for protection against the wildtype strain of COVID-19 versus a cutoff value of 4159 AU/mL for the Omicron variant in a cohort of dialysis patients using the identical microparticle immunoassay used in our study [43]. This difference in receiver operating curve (ROC) analyses indicates that immunogenicity conferred by ChAdOX1 and BNT162b2 is less effective at neutralizing the Omicron variant. Antibody levels in all cases of severe and fatal disease exceeded 296 AU/mL, but the timing of infection and disease severity suggest that these patients were infected with the Omicron variant rather than the wildtype strain of SARS-CoV-2. In conclusion, four doses of the COVID-19 vaccine induced a significantly stronger humoral response than those receiving three doses in COVID-19-naïve HD patients. Although the fourth COVID-19 vaccine has a partial protective role in reducing disease severity, decreased efficacy against evolved strains of the virus necessitates strain-specific protection strategy.

This study can be meaningful in that it serially observed the changes in humoral antibodies throughout the vaccination course of HD patients and demonstrates differences in antibody response according to COVID-19 infection status and severity. One of the limitations of this study is the small overall sample size and small number of severe cases, which led to low statistical power and generalizability of results. Obtaining more frequent spike and neutralizing antibody measures would be helpful in analyzing antibody kinetics and evaluating an accurate durability of protection. In addition, it was difficult to accurately assess the severity of all infections because most patients with confirmed infections were transferred to designated dialysis centers during the pandemic. Further studies with a larger cohort and more severe COVID-19 cases, as well as more granular measurement of antibody levels, will aid to complement the results of this study.

## Figures and Tables

**Figure 1 vaccines-11-01584-f001:**
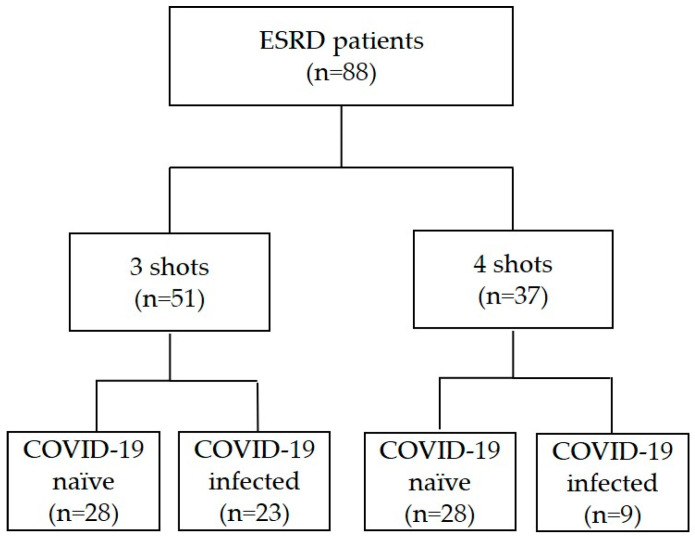
Flow diagram of study population.

**Figure 2 vaccines-11-01584-f002:**
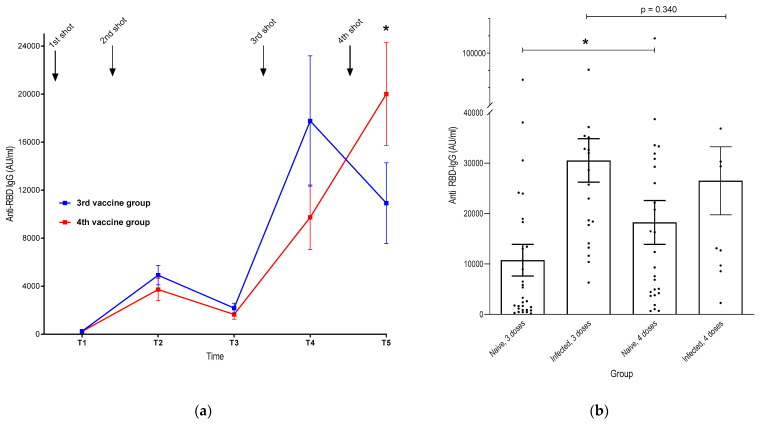
Comparison of anti-RBD IgG levels in: (**a**) COVID-19-naïve patients who received three versus four doses of vaccine over time; (**b**) for all patients at T5. Asterisk (*) indicates *p* < 0.05.

**Figure 3 vaccines-11-01584-f003:**
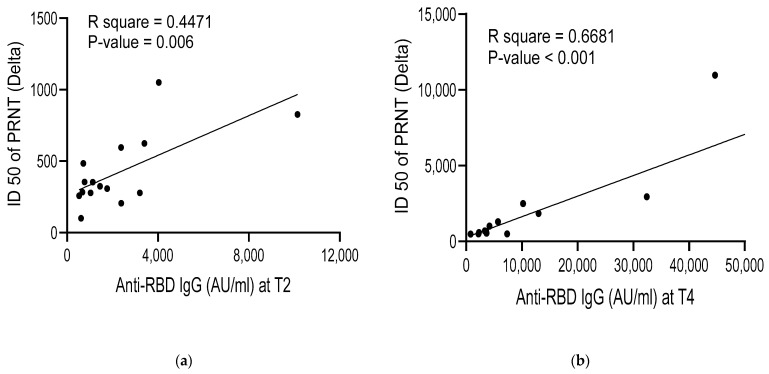
Correlation between anti-RBD IgG and neutralizing antibody at: (**a**) T2 with Delta variant; (**b**) T4 with Delta variant; (**c**) T2 with Omicron variant; (**d**) T4 with Omicron variant.

**Figure 4 vaccines-11-01584-f004:**
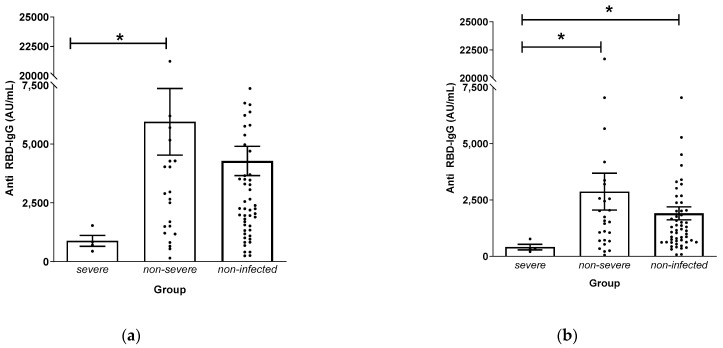
Comparison of antibody levels in severe, non-severe, and non-infected patients at: (**a**) T2; (**b**) T3. Asterisk (*) indicates *p* < 0.05.

**Figure 5 vaccines-11-01584-f005:**
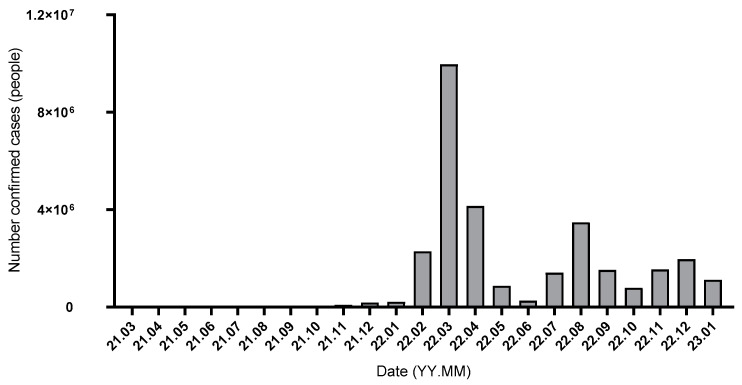
Trend of COVID-19 infection in South Korea.

**Table 1 vaccines-11-01584-t001:** Timeline of patient vaccination and antibody measurements.

Date	Event	Definition
30 April 2021	1st vaccination	
6–7 July 2021	Anti-RBD ^1^ IgG measurement	T1
13–16 July 2021	2nd vaccination	
6–7 September 2021	Anti-RBD IgG measurement PRNT ^2^	T2
8 November 2021	Anti-RBD IgG measurement	T3
22 November~23 December 2021	3rd vaccination	
28 March 2022	Anti-RBD IgG measurement PRNT	T4
25 April~8 November 2022	4th vaccination	
7 December 2022	Anti-RBD IgG measurement	T5

^1^ Antibody for the receptor binding domain (RBD) of COVID-19, ^2^ Plaque reduction neutralization test.

**Table 2 vaccines-11-01584-t002:** Baseline characteristics of the study population.

Parameter	Total (*n* = 88)	Naïve, 3 Doses (*n* = 28)	Infected, 3 Doses (*n* = 23)	Naïve, 4 Doses (*n* = 28)	Infected, 4 Doses (*n* = 9)	*p*-Value
Age (yr)	59.1 ± 10.6	55.0 ± 8.9	57.5 ± 13.7	64.7 ± 7.7	58.6 ± 8.2	0.005
Male, *n* (%)	56 (63.6)	14 (50.0)	14 (60.9)	20 (71.4)	8 (88.9)	0.136
BMI (kg/m^2^)	23.7 ± 4.7	23.0 ± 4.2	26.2 ± 5.9	22.4 ± 3.4	22.9 ± 4.2	0.019
Dialysis						
vintage (mo)	76.3 ± 54.8	83.9 ± 57.8	76.7 ± 54.9	73.6 ± 53.5	60.3 ± 54.0	0.719
Hemoglobin (g/dL)	10.9 ± 1.0	10.9 ± 1.0	11.1 ± 1.0	10.8 ± 1.0	10.7 ± 1.2	0.722
Albumin (g/dL)	4.1 ± 0.8	4.0 ± 0.3	3.9 ± 0.3	4.2 ± 1.3	4.1 ± 0.3	0.692
Cholesterol (mg/dL)	129.8 ± 29.8	134.7 ± 33.7	127.1 ± 31.1	125.2 ± 26.8	135.1 ± 22.4	0.606
hs-CRP ^1^ (mg/L)	1.1 ± 3.6	1.1 ± 2.0	1.7 ± 6.4	0.8 ± 1.4	0.9 ± 0.8	0.813
Kt/V	1.7 ± 0.3	1.7 ± 0.3	1.6 ± 0.3	1.7 ± 0.4	1.6 ± 0.2	0.342
Intact PTH ^2^ (pmol/L)	409.3 ± 362.9	492.2 ± 457.1	442.5 ± 372.6	352.2 ± 280.5	253.5 ± 154.7	0.269
Hemoglobin A1c ^3^ (%)	6.3 ± 1.3	6.1 ± 0.9	6.4 ± 1.5	5.9 ± 1.1	7.5 ± 1.5	0.071

^1^ High-sensitivity C-reactive protein (hs-CRP), ^2^ Parathyroid hormone (PTH), ^3^ Hemoglobin A1c was measured only in patients with diabetes.

**Table 3 vaccines-11-01584-t003:** Serum anti-RBD IgG levels according to vaccination status and COVID-19 infection status.

Anti-RBD IgG Levels (AU/mL)	Total (*n* = 88)	Naïve, 3 Doses (*n* = 28)	Infected, 3 Doses (*n* = 23)	Naïve, 4 Doses (*n* = 28)	Infected, 4 Doses (*n* = 9)	*p* Value
T1	83.3 (37.6–176.6)	129.7 (51.5–296.3)	87.1 (34.4–212.7)	75.1 (44.4–156.2)	23.6 (14.8/127.3)	0.064
T2	2777.6 (381.1–6064.3)	3485.0 (1915.6–6727.7)	2966.1 (1174.6–7598.1)	2220.6 (1031.8–4906.5)	2897.0 (1080.2–4155.3)	0.360
T3	1255.7 (640.2–2433.4)	1438.0 (894.0–2934.2)	1523.6 (672.7–3201.9)	806.0 (484.2–2009.9)	1049.8 (391.9–2067.3)	0.304
T4	5148.0 (2193.5–31,096.1)	4288.2 (2510.0–25,827.4)	29,537.7 (3309.3–81,815.4)	2758.6 (1523.9–9752.2)	4402.7 (2819.0–30,290.6)	0.039 ^1^
T5	13,221.9 (3717.74–31,375.2)	2492.5 (851.2–17,086.3)	28,883.4 (14,987.0–36,717.6)	14,075.3 (4527.1–30,460.0)	13,107.5 (9110.3–48,038.3)	0.000 ^2^

^1^ Post hoc analysis did not show a difference between groups, ^2^ post hoc analysis showed a significant difference only between naïve, 3 doses and naïve, 4 doses (*p* = 0.006).

**Table 4 vaccines-11-01584-t004:** Correlation between covariates and anti-RBD IgG values.

Co− Variates	Time				
	T1	T2	T3	T4	T5
Age ^1^ (years)	−0.007 (0.951)	−0.162 (0.131)	−0.113 (0.293)	−0.104 (0.341)	0.212 (0.051)
BMI ^1^ (kg/m^2^)	0.002 (0.984)	−0.159 (0.140)	−0.079 (0.464)	0.072 (0.515)	0.199 (0.069)
Dialysis vintage ^1^ (months)	0.033 (0.757)	0.116 (0.284)	0.059 (0.584)	0.073 (0.506)	−0.080 (0.465)
Sex ^2^	0.314 (0.003)	0.071 (0.511)	0.012 (0.909)	0.049 (0.657)	−0.048 (0.664)
Diabetes ^2^	−0.063 (0.559)	0.083 (0.440)	0.105 (0.332)	−0.012 (0.911)	−0.061 (0.579)

^1^ Continuous variables were analyzed using Spearman’s correlation. Values are presented as rho (*p* value), ^2^ categorical variables were analyzed using univariate linear regression. Values are presented as standardized beta (*p* value).

**Table 5 vaccines-11-01584-t005:** Characteristics of patients with severe COVID-19 disease.

Description	Date of Vaccination ^1^ (3rd Dose)	Date of Infection ^1^	Date of Death (If Applicable) ^1^	Antibody Level Prior to Infection (AU/mL)
72-year-old male patient with hypertension on dialysis for 4 years	2021-12-02	2022-09-19	N/A	553.2
61-year-old male patient with hypertension on dialysis for 8 years	2021-12-14	2022-07-21	N/A	788.8
59-year-old female patient with diabetes on dialysis for 16.8 years	2021-12-08	2022-03-11	2022-03-22	342
69-year-old female patient with diabetes on dialysis for 11.5 years	2021-12-03	2022-03-15	2022-03-28	317

^1^ All dates are expressed as YYYY-MM-DD.

## Data Availability

The data presented in this study are available on request from the corresponding author. The data are not publicly available due to identifying information such as age, sex, co-morbidities, and date of vaccination.

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
