# Peer review of "Comparison of Humoral Response between Third and Fourth Doses of COVID-19 Vaccine in Hemodialysis Patients"

_vaccines, 2023, doi:10.3390/vaccines11101584_

Round 1
Reviewer 1 Report
This manuscript offers a novel and impactful contribution by investigating the humoral response to COVID-19 vaccination and its relationship to disease severity in a unique and vulnerable cohort—patients with end-stage renal disease undergoing hemodialysis. Notably, it evaluates the efficacy of a fourth COVID-19 vaccine dose, a timely and pertinent inquiry when booster recommendations were less established. The study's findings shed light on the importance of robust antibody responses in protecting against severe COVID-19 outcomes in this specific population, which is known to have compromised immune responses. However, the manuscript has some shortcomings which are highlighted below:
The manuscript exhibits several notable grammatical errors, which impede its readability and comprehension. Complex sentence structures often hinder clarity, and subject-verb agreement issues are prevalent throughout the manuscript.
The abstract mentions measuring humoral response at five different times and plaque reduction neutralization tests at two different times after vaccination. However, the duration of follow-up is not provided. The abstract does not mention whether other potential confounding factors (e.g., age, comorbidities, medications) were considered in the analysis. The abstract mentions that severe COVID-19 infection is defined as hospital admission or death. This definition may not capture the full spectrum of severe disease, as some patients with severe symptoms may not be hospitalized or may recover without hospitalization.
The introduction mentions that the etiology of increased mortality in ESRD patients with COVID-19 is not fully understood and briefly discusses impaired immune responses in these patients. However, it does not delve into the complexities of this issue or provide a comprehensive overview of the existing research on the topic. A more detailed discussion of the potential factors contributing to increased susceptibility to COVID-19 in ESRD patients would enhance the introduction. While the introduction mentions "in vivo study of patients with ESRD," it does not specify the sources of data or provide details about the study design. Transparency about the data sources and study methodology is crucial for assessing the reliability of the information presented. The introduction mentions the utility of a third SARS-CoV-2 vaccine but states that there is insufficient evidence to validate the need for a fourth or fifth booster dose. However, it does not clarify the criteria or studies used to determine this lack of evidence. Providing a more comprehensive overview of the current state of research on booster doses in ESRD patients would be informative. The introduction lacks a clear statement of the study's specific objectives and research questions. It is important to clearly define what the study aims to investigate and why it is relevant.
The sample size in this study is relatively small, with a total of 88 patients. A larger sample size would provide more statistical power and improve the generalizability of the findings, especially when analyzing subgroups. Additionally, the method of patient selection and potential biases related to this selection process are not fully addressed. While the study measures anti-RBD IgG levels at multiple time points, it would be valuable to have more frequent measurements, especially during the critical post-vaccination period. A more granular assessment of antibody response kinetics could provide insights into the dynamics of immune response. The methodology defines severe disease as hospital admission or death, but it does not provide details on how hospitalization criteria were determined. Clear and standardized criteria for hospital admission would enhance the study's reliability and reproducibility.
The results section mentions only four patients classified as having severe COVID-19 disease, with two fatal cases and two requiring hospitalization. Such a small number of severe cases makes it challenging to draw statistically meaningful conclusions about the relationship between antibody levels and disease severity. Additionally, the absence of statistical analysis to support the observed differences in antibody levels is a limitation. The section discusses the antibody levels in severe cases prior to infection and compares them to non-severe patients. However, the analysis lacks statistical tests or measures of effect size to determine whether the observed differences are statistically significant or clinically meaningful. The results section does not mention the inclusion of a control group for comparison. A control group, consisting of dialysis patients who did not contract COVID-19, would be essential for assessing the impact of vaccination and antibody levels on disease severity. The results do not appear to include multivariate analysis to account for potential confounding factors that could influence disease severity. Variables such as age, comorbidities, and other patient characteristics should be considered in the analysis.
The discussion places significant emphasis on age as a factor influencing vaccination and infection rates. While age can be an important factor, it should be discussed in the context of other potential variables and controlled for in the analysis. Additionally, the discussion should explore other potential factors contributing to vaccination decisions. The discussion briefly mentions the relationship between antibody levels and disease severity but does not delve into the clinical significance of specific antibody levels. More detailed interpretation of what antibody levels signify in terms of protection or susceptibility to COVID-19 is needed. While the discussion mentions the analysis of neutralizing antibodies, it does not provide a comprehensive discussion of the results or their implications. The significance of neutralizing antibodies in protection against variants should be explored in greater detail. The discussion could benefit from a paragraph outlining recommendations for future research. This could include suggestions for larger studies, additional variables to consider, and the need for long-term follow-up to assess the durability of antibody responses. I believe addressing these points will improve the clarity and impact of the manuscript.
Moderate editing of English language is required.
Author Response
|
Response to Reviewer 1 Comments
|
||
|
1. Summary |
|
|
|
Thank you very much for your insightful feedback regarding our submission. We have addressed the comments and suggestions as best as possible, so please find the detailed responses below and the corresponding revisions/corrections highlighted in the re-submitted files. |
||
|
2. Questions for General Evaluation |
Reviewer’s Evaluation |
Response and Revisions |
|
Does the introduction provide sufficient background and include all relevant references? |
Yes/Can be improved/Must be improved/Not applicable |
|
|
Are all the cited references relevant to the research? |
Yes/Can be improved/Must be improved/Not applicable |
|
|
Is the research design appropriate? |
Yes/Can be improved/Must be improved/Not applicable |
|
|
Are the methods adequately described? |
Yes/Can be improved/Must be improved/Not applicable |
|
|
Are the results clearly presented? |
Yes/Can be improved/Must be improved/Not applicable |
|
|
Are the conclusions supported by the results? |
Yes/Can be improved/Must be improved/Not applicable |
|
|
3. Point-by-point response to Comments and Suggestions for Authors |
||
|
Comments 1: The manuscript exhibits several notable grammatical errors, which impede its readability and comprehension. Complex sentence structures often hinder clarity, and subject-verb agreement issues are prevalent throughout the manuscript. |
||
|
Response 1: Thank you for your comment. We have revised grammatical errors and complex sentence structures to enhance clarity. |
||
|
Comments 2: The abstract mentions measuring humoral response at five different times and plaque reduction neutralization tests at two different times after vaccination. However, the duration of follow-up is not provided. |
||
|
Response 2: We have revised the abstract to specify the duration of study and the time points at which antibody and/or plaque reduction neutralization tests were performed. The study follow-up period was 18 months. The mentioned change can be found on page 1, paragraph 1, lines 28-31 of the revised manuscript and is highlighted as the following text: “Antibody levels were measured at approximately two-month intervals after the first and second dose, then four months after the third dose, then one to six months after the fourth dose of vaccine. PRNT was performed two months after the second and four months after the third dose of vaccine.” |
||
|
Comments 3: The abstract does not mention whether other potential confounding factors (e.g., age, comorbidities, medications) were considered in the analysis. |
||
|
Response 3: We analyzed age, body mass index (BMI), sex, dialysis vintage, and presence of diabetes mellitus as potential confounding factors. Analyses did not show any consistent and statistically significant correlation between clinically important covariates and antibody levels. The methods, result, and discussion of our analysis have been revised in the corresponding sections. We have also revised the highlighted text of the abstract on page 1, paragraph 1, lines 35-36 to reflect our findings as the following statement: “Age, gender, body mass index (BMI), dialysis vintage, and presence of diabetes mellitus (DM) did not show a significant correlation with antibody levels.” |
||
|
Comments 4: The abstract mentions that severe COVID-19 infection is defined as hospital admission or death. This definition may not capture the full spectrum of severe disease, as some patients with severe symptoms may not be hospitalized or may recover without hospitalization. |
||
|
Response 4: The definition of severe COVID-19 has been revised on page 1, paragraph 1, lines 32-33 as “hospital admission for greater than or equal to two weeks” rather than simply “hospital admission”. Our definition of severe disease was used to reflect disease severity by including patients who required prolonged admission (longer than the mandatory quarantine period as required by the Korean government) due to complications of COVID-19. |
||
|
Comments 5: The introduction mentions that the etiology of increased mortality in ESRD patients with COVID-19 is not fully understood and briefly discusses impaired immune responses in these patients. However, it does not delve into the complexities of this issue or provide a comprehensive overview of the existing research on the topic. A more detailed discussion of the potential factors contributing to increased susceptibility to COVID-19 in ESRD patients would enhance the introduction. |
||
|
Response 5: Thank you for your feedback. I agree that a more comprehensive overview will enhance the introduction, and have made appropriate changes in pages 1-2, paragraph 1, lines 46-59. We have elaborated on the various processes by which dialysis patients are more susceptible to infection, ranging from cellular level (i.e. immune cell function) to systemic issues (i.e. dialysis visits and increased exposure). We also noted that dialysis patients tend to be older and have more co-morbidities that contribute to immune impairment compared to the general population. |
||
|
Comments 6: While the introduction mentions "in vivo study of patients with ESRD," it does not specify the sources of data or provide details about the study design. Transparency about the data sources and study methodology is crucial for assessing the reliability of the information presented. |
||
|
Response 6: We have deleted the statement because we felt that changes made in response 5 were sufficient to explain the mechanism of immune impairment in dialysis patients. |
||
|
Comments 7: The introduction mentions the utility of a third SARS-CoV-2 vaccine but states that there is insufficient evidence to validate the need for a fourth or fifth booster dose. However, it does not clarify the criteria or studies used to determine this lack of evidence. Providing a more comprehensive overview of the current state of research on booster doses in ESRD patients would be informative. |
||
|
Response 7: The study period for this manuscript lasted from April 2021 to December 2022. Approval for a third booster vaccination was not released until September 2021 (in the United States), and first clinical data regarding efficacy of third vaccine were not published until October 20211. Therefore, in the context of this study, there was insufficient evidence to recommend a fourth booster vaccination to patients during the study period. With more recent studies examining patients who received four vaccinations, we recognize that stating there is “insufficient evidence to validate the need for a fourth or fifth booster dose” may be antiquated. Therefore, we have supplemented the introduction with additional references that have examined the efficacy and cost-effectiveness of a third booster vaccine in page 2, paragraph 3, line 72-74 of the manuscript, and have added additional reference data from more recent studies that have examined the efficacy of a fourth COVID vaccine on page 2, paragraph 3, lines 75-79. The abstract has also been rephased on page 1, paragraph 1, lines 21-23 as “…only a few studies have evaluated the efficacy of a fourth booter vaccination in hemodialysis (HD) patients and there is not enough evidence to recommend for or against a fourth booster vaccination.”
1Reference : Miller, Jacqueline M. (2021). Safety and immunogenicity of a 50 µg booster dose of Moderna COVID-19 vaccine. 202113(120203). |
||
|
Comments 8: The introduction lacks a clear statement of the study's specific objectives and research questions. It is important to clearly define what the study aims to investigate and why it is relevant. |
||
|
Response 8: In light of response 7, we have rephrased the objective of the study in page 2, paragraph 3, lines 80-83 of the introduction as the following: “…we examined the effect of a fourth SARS-CoV-2 vaccine for boosting antibody response in HD patients…by (1) serial measurement of antibody levels and neutralizing antibody and (2) comparison of patients who received either three or four doses of vaccine, along with COVID-19 infection status.” We believe two important factors differentiate our study from pre-existing studies. First, we classified patients based on the presence of COVID-19 infection and the number of vaccine doses. While other studies indicate that previous COVID-19 infection may alter antibody kinetics after booster vaccination1, our classification allows for a more specific comparison between number of vaccinations and disease severity. Second, we serially examined antibody levels in ESRD patients after the first, second, and third vaccine doses. Antibody response to previous vaccination was a factor that was not considered in most studies since studies that examined the effect of booster vaccines started with the premise that patients completed (but may or may not have responded to) the recommend two-dose vaccine series2.
1Reference: Rodda LB, Morawski PA, Pruner KB, Fahning ML, Howard CA, Franko N, Logue J, Eggenberger J, Stokes C, Golez I, Hale M. Imprinted SARS-CoV-2-specific memory lymphocytes define hybrid immunity. Cell. 2022 Apr 28;185(9):1588-601. 2Reference: Cohen-Hagai, K., Hornik-Lurie, T., Benchetrit, S. et al. Clinical efficacy of the fourth dose of the BNT162b2 vaccine in maintenance dialysis patients . J Nephrol 36, 1957–1964 (2023). https://doi.org/10.1007/s40620-023-01667-z |
||
|
Comments 9: The sample size in this study is relatively small, with a total of 88 patients. A larger sample size would provide more statistical power and improve the generalizability of the findings, especially when analyzing subgroups. |
||
|
Response 9: This is a valid comment, and we strongly agree that a larger sample size would provide more statistical power and generalizability. However, we only enrolled patients with follow-up data from two dialysis centers and have acknowledged small sample size as one of the limitations of our study on page 11, paragraph 2, lines 324-326 of the manuscript. |
||
|
Comments 10: Additionally, the method of patient selection and potential biases related to this selection process are not fully addressed. |
||
|
Response 10: Inclusion and exclusion criteria are added to page 2, paragraph 4, lines 86-89 as the following: “Enrolled patients were between ages 18 to 90 and receiving maintenance dialysis for greater than three months. Patients with a life expectancy of less than six months, active malignancy, and those who refused vaccination were excluded from the study”. |
||
|
Comments 11: While the study measures anti-RBD IgG levels at multiple time points, it would be valuable to have more frequent measurements, especially during the critical post-vaccination period. A more granular assessment of antibody response kinetics could provide insights into the dynamics of immune response. |
||
|
Response 11: We agree with the reviewer’s comment that a more granular assessment of antibody levels would show more information about antibody response kinetics. While we were able to obtain relatively regular antibody measures after the first, second, and third vaccines, there was not much evidence or consensus for a fourth vaccination during the study period. Hence, the acceptance and timing of a fourth vaccine was strictly voluntary. Because patients received vaccination over period of six months, it was not possible to obtain consistent, regular measurements of antibody levels. |
||
|
Comments 12: The methodology defines severe disease as hospital admission or death, but it does not provide details on how hospitalization criteria were determined. Clear and standardized criteria for hospital admission would enhance the study's reliability and reproducibility. |
||
|
Response 12: We have edited the definition of severe disease on page 3, paragraph 2, line 133 as “hospital admission for greater than two weeks” instead of simply “hospital admission”. Different studies define “severe” COVID-19 in different ways, including hospital admission, oxygen requirement, and death1,2,3. The national policy for COVID-19 positive hemodialysis patients in Korea included (1) transfer to designed hemodialysis centers (where only infected patients received dialysis in cohort) and (2) mandatory quarantine for one to two weeks. Because patients were transferred to another hospital during infection, it was not possible to obtain accurate information about disease course, especially oxygen requirement. Our definition of severe disease was used to reflect disease severity by including patients who required prolonged admission (longer than the mandatory quarantine period) due to complications of COVID-19.
1 Reference: Ashby, Damien R.; Caplin, Ben; Corbett, Richard W.; Asgari, Elham; Kumar, Nicola; Sarnowski, Alexander; Hull, Richard; Makanjuola, David; Cole, Nicholas; Chen, Jian; Nyberg, Sofia; McCafferty, Kieran; Zaman, Faryal; Cairns, Hugh; Sharpe, Claire; Bramham, Kate; Motallebzadeh, Reza; Anwari, Kashif Jamil; Salama, Alan D.; Banerjee, Debasish* on behalf of the Pan-London COVID-19 Renal Audit Group. Severity of COVID-19 after Vaccination among Hemodialysis Patients: An Observational Cohort Study. CJASN 17(6):p 843-850, June 2022. | DOI: 10.2215/CJN.16621221 2 Reference: Wing, Sara; Thomas, Doneal; Balamchi, Shabnam; Ip, Jane; Naylor, Kyla; Dixon, Stephanie N.; McArthur, Eric; Kwong, Jeffrey C.; Perl, Jeffrey; Atiquzzaman, Mohammad; Yeung, Angie; Yau, Kevin; Hladunewich, Michelle A.; Leis, Jerome A.; Levin, Adeera; Blake, Peter G.; Oliver, Matthew J.. Effectiveness of Three Doses of mRNA COVID-19 Vaccines in the Hemodialysis Population during the Omicron Period. Clinical Journal of the American Society of Nephrology 18(4):p 491-498, April 2023. | DOI: 10.2215/CJN.0000000000000108 3 Reference: Sakhi H, Dahmane D, Attias P, et al. Kinetics of Anti-SARS-CoV-2 IgG Antibodies in Hemodialysis Patients Six Months after Infection. J Am Soc Nephrol. 2021;32(5):1033-1036. doi:10.1681/ASN.2020111618 |
||
|
Comments 13: The results section mentions only four patients classified as having severe COVID-19 disease, with two fatal cases and two requiring hospitalization. Such a small number of severe cases makes it challenging to draw statistically meaningful conclusions about the relationship between antibody levels and disease severity. Additionally, the absence of statistical analysis to support the observed differences in antibody levels is a limitation. The section discusses the antibody levels in severe cases prior to infection and compares them to non-severe patients. However, the analysis lacks statistical tests or measures of effect size to determine whether the observed differences are statistically significant or clinically meaningful. The results section does not mention the inclusion of a control group for comparison. A control group, consisting of dialysis patients who did not contract COVID-19, would be essential for assessing the impact of vaccination and antibody levels on disease severity. |
||
|
Response 13: We agree with the reviewer’s comment and would like to thank him or her for their insightful feedback. A small number of severe cases is an inherent weakness that follows suit with our small sample size. For further statistical analyses, we have added Figure 4a and 4b, which compares the average antibody level in severe (infected), non-severe (infected) and non-infected patients using Kruskal-Wallis test. Data was only available for T2 and T3 measurements because two deceased patients were censored after T3. A description of statistical analyses were added in the materials section (page 4, paragraph 3, lines 152-154). We have also added a descriptive statement on page 8, paragraph 1, lines 211-217 to explain the results of Figure 4. It is questionable whether non-infected patients are an accurate control group for comparison of disease severity since non-infection does not necessarily guarantee superior antibody formation. As noted on page 10, paragraph 1, lines 258-260, severe and fatal cases were not different from the rest of the study population in terms of age or the use of immunosuppressants, which can potentially impair antibody formation. Our previous rationale in comparing only severe and non-severe patients was that non-infected patients are governed by more confounding variables that affect their likelihood of acquiring COVID-19. Nevertheless, we would like to thank the reviewer for mentioning the need of a non-infected control group, as statistical analysis did show a significant difference. |
||
|
Comments 14: The results do not appear to include multivariate analysis to account for potential confounding factors that could influence disease severity. Variables such as age, comorbidities, and other patient characteristics should be considered in the analysis. The discussion places significant emphasis on age as a factor influencing vaccination and infection rates. While age can be an important factor, it should be discussed in the context of other potential variables and controlled for in the analysis. |
||
|
Response 14: We performed a correlation analysis using the following variables: age, BMI, dialysis vintage, sex, and presence of diabetes mellitus and have reported our findings on Table 4. Analyses did not show any consistent and statistically significant correlation between clinically important covariates and antibody levels. Description of our data has been edited on page 5, paragraph 1, lines 180-183. We have also discussed age and the lack of correlation between age and antibody formation to a further extent on page 9, paragraph 1, lines 240-246, indicating that small sample size and natural immunity following COVID-19 infection may act as potential confounding factors. |
||
|
Comments 15: Additionally, the discussion should explore other potential factors contributing to vaccination decisions. |
||
|
Response 15: We have edited page 9, paragraph 1, lines 233-236 to discuss other potential factors contributing to vaccination decisions with addition of the following sentence: “Patient reluctance toward the fourth vaccine was multi-faceted, owing to vaccine fatigue, suspicion about vaccine reliability, and mass media coverage reporting Omicron variant as being less deadly than previous variants.” |
||
|
Comments 16: The discussion briefly mentions the relationship between antibody levels and disease severity but does not delve into the clinical significance of specific antibody levels. More detailed interpretation of what antibody levels signify in terms of protection or susceptibility to COVID-19 is needed. |
||
|
Response 16: We have elaborated on the significance of antibody levels and cutoff values with the addition on page 10-11, paragraph 4, lines 308-316. Currently, there is no established definition for a cut-off level that confers seroprotection. The cited reference by Affeldt et al1 analyzed serum anti-RBD IgG levels using the same microparticle immunoassay used in our study in a cohort of hemodialysis patients. The authors suggested a cut-off value of 296 AU/mL for the wildtype strain, and 4159 AU/mL for the Omicron variant of SARS-CoV-2 to elicit a meaningful neutralizing activity. We point out that cases of severe and fatal COVID-19 showed an antibody level that exceeded 296 AU/mL but was smaller than 4159 AU/mL. Low antibody levels, along with time frame of infection during with Omicron was the dominant variant, hints that low antibody levels in these patients were not protective enough against severe Omicron infection.
1 Reference: Affeldt, P.; Koehler, F.C.; Brensing, K.A.; Gies, M.; Platen, E.; Adam, V.; Butt, L.; Grundmann, F.; Heger, E.; Hinrichs, S.; et al. Immune Response to Third and Fourth COVID-19 Vaccination in Hemodialysis Patients and Kidney Transplant Recipients. Viruses 2022, 14, 2646. https://doi.org/10.3390/v14122646 |
||
|
Comments 17: While the discussion mentions the analysis of neutralizing antibodies, it does not provide a comprehensive discussion of the results or their implications. The significance of neutralizing antibodies in protection against variants should be explored in greater detail. |
||
|
Response 17: A discussion of the results of plaque reduction neutralization test has been augmented. We noted that the significant degree of correlation between serum anti-RBD IgG and neutralizing antibody indicates good protection against Delta and Omicron variants. For Omicron variant at T2, which did not show a significant correlation, we hypothesized that lower anti-RBD IgG level is a potential reason for lack of statistical significance, since antibody levels at T4 show a significant correlation (R2 = 0.786). We have also reviewed the differences in neutralization capacity between infection and vaccination. Changes can be found on page 10, paragraph 3, lines 295-298, 301-302, 306-307. |
||
|
Comments 18: The discussion could benefit from a paragraph outlining recommendations for future research. This could include suggestions for larger studies, additional variables to consider, and the need for long-term follow-up to assess the durability of antibody responses. |
||
|
Response 18: A paragraph outlining limitations and areas of future study has been added on page 11, paragraph 2, lines 322-332. The limitations of this study include (1) small sample size, (2) small number of severe cases, (3) infrequent antibody measurements, (4) incomplete description of disease course in COVID-19 infected patients. Areas of future study include larger studies with more severe COVID-19 cases, as well as more frequent measurements of antibody levels. |
||
|
4. Response to Comments on the Quality of English Language |
||
|
Point 1: Moderate editing of English language is required. |
||
|
Response 1: We have revised grammatical errors and complex sentence structures to enhance clarity. |
||
|
5. Additional clarifications |
||
|
On page 3, paragraph 2, line 116, the definition of T1 has been revised from 1 month to 2 month after the first dose of COVID-19 vaccine because the time interval between April 30 and July 6 is 68 days, which is approximately two months.
We deleted column 6 of Table 5 because the average value of non-severe cases was identical for all patients and have replaced it with descriptive data on page 8, paragraph 1, lines 211-212. Stating “the mean value in non-severe COVID-19 patients was 2870.7 AU/mL and 1905.6 AU/mL for non-infected patients.” The contents of row 2 have been replaced from “69-year-old male patient with diabetes on dialysis for 3 years and liver transplant recipient due to chronic hepatitis B and cirrhosis” to “72-year-old male with hypertension on dialysis for 4 years,” due to an honest error. The date of vaccination, date of infection, and antibody level prior to infection have been changed to December 2, 2021, September 19, 2022, and 553.2 AU/mL, respectively.
Table 4. Correlation between covariates and anti-RBD IgG values has been added to page 6 of the manuscript to display the results of correlation analyses. All tables following table 4 have been re-numbered.
On page 7, Figure 3, data were previously expressed as anti-RBD IgG at T2 and T3. This is a mistake and does not correspond to the results shown in Table 1. Axis labels and caption were revised to display the correct time of testing (T2 and T4).
Figure 4. Comparison of antibody levels in severe, non-severe, and non-infected patients has been added to page 8-9 of the manuscript to provide statistical analyses comparing these patients. All figures following figure 4 have been re-numbered.
Combining the results of sequential antibody measurement and observation of severe patients, we have revised our discussion on page 10, paragraph 2, lines 272-282 to further explicate our conclusions. Sequential measurement of anti-RBD IgG shows that antibody levels fall at 3-6 months following vaccination. Observational data of severe COVID-19 cases, though not statistically significant, links severe disease with low antibody levels. Therefore, boosting antibody levels through repeated vaccination will protect against severe disease in hemodialysis patients infected with COVID-19. |
||
Reviewer 2 Report
The manuscript "Comparison of Humoral Response between Third and Fourth Doses of COVID-19 Vaccine in Hemodialysis Patients" explores the humoral response to COVID-19 vaccination in hemodialysis patients, particularly comparing those who received three doses of the vaccine to those who received four doses.
Overall the manuscript is well-written, and the objective of the study is well-defined
-The authors should mention the inclusion and exclusion criteria
-The sample size is small because the patients were divided into further subgroups. It will be good if authors can increase the sample size that will strengthen the study
-The manuscript lacks data on the duration of protection conferred by the third and fourth vaccine doses, which is essential for understanding the long-term benefits of additional doses
-Alignment issue in Figure 3: Align "Figure b" according to "figure a"
Author Response
|
Response to Reviewer 2 Comments
|
||
|
1. Summary |
|
|
|
Thank you very much for your insightful feedback regarding our submission. We have addressed the comments and suggestions as best as possible, so please find the detailed responses below and the corresponding revisions/corrections highlighted changes in the re-submitted files. |
||
|
2. Questions for General Evaluation |
Reviewer’s Evaluation |
Response and Revisions |
|
Does the introduction provide sufficient background and include all relevant references? |
Yes/Can be improved/Must be improved/Not applicable |
|
|
Are all the cited references relevant to the research? |
Yes/Can be improved/Must be improved/Not applicable |
|
|
Is the research design appropriate? |
Yes/Can be improved/Must be improved/Not applicable |
|
|
Are the methods adequately described? |
Yes/Can be improved/Must be improved/Not applicable |
|
|
Are the results clearly presented? |
Yes/Can be improved/Must be improved/Not applicable |
|
|
Are the conclusions supported by the results? |
Yes/Can be improved/Must be improved/Not applicable |
|
|
3. Point-by-point response to Comments and Suggestions for Authors |
||
|
Comments 1: The authors should mention the inclusion and exclusion criteria
|
||
|
Response 1: Inclusion and exclusion criteria are added to page 2, paragraph 4, lines 86-89 as the following: “Enrolled patients were between ages 18 to 90 and receiving maintenance dialysis for greater than three months. Patients with a life expectancy of less than six months, active malignancy, and those who refused vaccination were excluded from the study”. |
||
|
Comments 2: The sample size is small because the patients were divided into further subgroups. It will be good if authors can increase the sample size that will strengthen the study |
||
|
Response 2: This is a valid comment, and we strongly agree that a larger sample size would provide more statistical power and generalizability. However, we only enrolled patients with follow-up data from two dialysis centers, and have acknowledged small sample size as one of the limitations of our study on page 11, paragraph 2, lines 324-325 of the manuscript. Our patient classification is based on two reasons. First, the administration of the fourth COVID vaccine was strictly voluntary. Second, we hypothesized that antibody kinetics of infected and non-infected patients would be different, which is the reason why divided patients into four categories. Current findings support our hypothesis1, so although the sample size is small, we believe that the rationale for our analysis is still reasonable.
1Reference: Rodda LB, Morawski PA, Pruner KB, Fahning ML, Howard CA, Franko N, Logue J, Eggenberger J, Stokes C, Golez I, Hale M. Imprinted SARS-CoV-2-specific memory lymphocytes define hybrid immunity. Cell. 2022 Apr 28;185(9):1588-601. |
||
|
Comments 3: The manuscript lacks data on the duration of protection conferred by the third and fourth vaccine doses, which is essential for understanding the long-term benefits of additional doses |
||
|
Response 3: In hemodialysis patients, antibody levels drop dramatically at approximately six months after vaccination1. Our data shows that a fourth booster vaccination increases antibody levels. Although more frequent data collection regarding antibody levels would have provided additional information about duration of protection, our main argument is that boosting antibody response through additional vaccination confers protection because based on serial data, patients with poor antibody response showed a tendency towards severe disease.
1Reference: Louise Füessl, Tobias Lau, Simon Rau, Ron Regenauer, Michael Paal, Sandra Hasmann, Florian M Arend, Mathias Bruegel, Daniel Teupser, Michael Fischereder, Ulf Schönermarck, Humoral response after SARS-CoV-2 booster vaccination in haemodialysis patients with and without prior infection, Clinical Kidney Journal, Volume 15, Issue 8, August 2022, Pages 1633–1635, https://doi.org/10.1093/ckj/sfac148 |
||
|
Comments 4: Alignment issue in Figure 3: Align "Figure b" according to "figure a" |
||
|
Response 4: We have adjusted the spacing between figures to align figure b with figure a. |
||
|
4. Response to Comments on the Quality of English Language |
||
|
5. Additional clarifications |
||
|
On page 3, paragraph 2, line 116, the definition of T1 has been revised from 1 month to 2 month after the first dose of COVID-19 vaccine because the time interval between April 30 and July 6 is 68 days, which is approximately two months.
We deleted column 6 of Table 5 because the average value of non-severe cases was identical for all patients and have replaced it with descriptive data on page 8, paragraph 1, lines 211-212. Stating “the mean value in non-severe COVID-19 patients was 2870.7 AU/mL and 1905.6 AU/mL for non-infected patients.” The contents of row 2 have been replaced from “69-year-old male patient with diabetes on dialysis for 3 years and liver transplant recipient due to chronic hepatitis B and cirrhosis” to “72-year-old male with hypertension on dialysis for 4 years,” due to an honest error. The date of vaccination, date of infection, and antibody level prior to infection have been changed to December 2, 2021, September 19, 2022, and 553.2 AU/mL, respectively.
Table 4. Correlation between covariates and anti-RBD IgG values has been added to page 6 of the manuscript to display the results of correlation analyses. All tables following table 4 have been re-numbered.
On page 7, Figure 3, data were previously expressed as anti-RBD IgG at T2 and T3. This is a mistake and does not correspond to the results shown in Table 1. Axis labels and caption were revised to display the correct time of testing (T2 and T4).
Figure 4. Comparison of antibody levels in severe, non-severe, and non-infected patients has been added to page 8-9 of the manuscript to provide statistical analyses comparing these patients. All figures following figure 4 have been re-numbered.
Combining the results of sequential antibody measurement and observation of severe patients, we have revised our discussion on page 10, paragraph 2, lines 272-282 to further explicate our conclusions. Sequential measurement of anti-RBD IgG shows that antibody levels fall at 3-6 months following vaccination. Observational data of severe COVID-19 cases, though not statistically significant, links severe disease with low antibody levels. Therefore, boosting antibody levels through repeated vaccination will protect against severe disease in hemodialysis patients infected with COVID-19. |
||
Reviewer 3 Report
A few minor comments and suggestions are included in the attached manuscript revision.

A few minor revisions suggested but no significant issues.
Author Response
|
Response to Reviewer 3 Comments
|
||
|
1. Summary |
|
|
|
Thank you very much for your feedback regarding our submission. We have addressed the comments and suggestions as best as possible, so please find the detailed responses below and the corresponding revisions/corrections highlighted changes in the re-submitted files. |
||
|
2. Questions for General Evaluation |
Reviewer’s Evaluation |
Response and Revisions |
|
Does the introduction provide sufficient background and include all relevant references? |
Yes/Can be improved/Must be improved/Not applicable |
|
|
Are all the cited references relevant to the research? |
Yes/Can be improved/Must be improved/Not applicable |
|
|
Is the research design appropriate? |
Yes/Can be improved/Must be improved/Not applicable |
|
|
Are the methods adequately described? |
Yes/Can be improved/Must be improved/Not applicable |
|
|
Are the results clearly presented? |
Yes/Can be improved/Must be improved/Not applicable |
|
|
Are the conclusions supported by the results? |
Yes/Can be improved/Must be improved/Not applicable |
|
|
3. Point-by-point response to Comments and Suggestions for Authors |
||
|
Comments 1: It looks like the authors classified patients as COVID-19-naive or -infected based upon their PCR tests during the study. Were any considerations given to prior infections with COVID-19 before the beginning of the study? Whether this was considered as a parameter should be discussed. |
||
|
Response 1: We realize that asymptomatic COVID-19 infection is a potential confounding factor in the analysis of COVID antibody. However, we considered this possibility to be low for the following reasons. First, the nationwide surveillance for COVID-19 was extremely thorough during the initial phase of the disease: all contacts of confirmed patients were tested regardless of symptoms, and no patients were confirmed positive during this period. Second, individual screening at dialysis center was extremely thorough. Body temperature and symptoms of upper respiratory infection were monitored before entering the dialysis clinic before every session, and symptomatic patients were not allowed to enter the facility without a negative PCR result. Third, COVID-19 PCR was performed in patients with even a low index of suspicion to prevent patient-to-patient transmission in dialysis clinics, making the possibility of asymptomatic infection unlikely. |
||
|
Comments 2: A few minor comments and suggestions are included in the attached manuscript revision: “unlikely.” |
||
|
Response 2: Thank you for your comment. We have accepted the reviewer’s suggestion and have edited page 10, paragraph 2, line 291 as “… there was nationwide surveillance for for fever and symptoms of upper respiratory tract infection, and COVID-19 PCR was performed in patients with a low index of suspicion, making the inclusion of asymptomatic infection unlikely.” |
||
|
Comments 3: A few minor comments and suggestions are included in the attached manuscript revision: “may” |
||
|
Response 3: The aforementioned sentence has been revised as the following on page 11, paragraph 1, lines 318-321: “Although the fourth COVID-19 vaccine has a partial protective role in reducing disease severity, decreased efficacy against evolved strains of the virus necessitates strain-specific protection strategy.” |
||
|
4. Response to Comments on the Quality of English Language |
||
|
Point 1: A few minor revisions suggested but no significant issues. |
||
|
. |
||
|
5. Additional clarifications |
||
|
On page 3, paragraph 2, line 116, the definition of T1 has been revised from 1 month to 2 month after the first dose of COVID-19 vaccine because the time interval between April 30 and July 6 is 68 days, which is approximately two months.
We deleted column 6 of Table 5 because the average value of non-severe cases was identical for all patients and have replaced it with descriptive data on page 8, paragraph 1, lines 211-212. Stating “the mean value in non-severe COVID-19 patients was 2870.7 AU/mL and 1905.6 AU/mL for non-infected patients.” The contents of row 2 have been replaced from “69-year-old male patient with diabetes on dialysis for 3 years and liver transplant recipient due to chronic hepatitis B and cirrhosis” to “72-year-old male with hypertension on dialysis for 4 years,” due to an honest error. The date of vaccination, date of infection, and antibody level prior to infection have been changed to December 2, 2021, September 19, 2022, and 553.2 AU/mL, respectively.
Table 4. Correlation between covariates and anti-RBD IgG values has been added to page 6 of the manuscript to display the results of correlation analyses. All tables following table 4 have been re-numbered.
On page 7, Figure 3, data were previously expressed as anti-RBD IgG at T2 and T3. This is a mistake and does not correspond to the results shown in Table 1. Axis labels and caption were revised to display the correct time of testing (T2 and T4).
Figure 4. Comparison of antibody levels in severe, non-severe, and non-infected patients has been added to page 8-9 of the manuscript to provide statistical analyses comparing these patients. All figures following figure 4 have been re-numbered.
Combining the results of sequential antibody measurement and observation of severe patients, we have revised our discussion on page 10, paragraph 2, lines 272-282 to further explicate our conclusions. Sequential measurement of anti-RBD IgG shows that antibody levels fall at 3-6 months following vaccination. Observational data of severe COVID-19 cases, though not statistically significant, links severe disease with low antibody levels. Therefore, boosting antibody levels through repeated vaccination will protect against severe disease in hemodialysis patients infected with COVID-19. |
||
Round 2
Reviewer 1 Report
I am impressed that the authors have substantially addressed the highlighted shortcomings. Now I feel confident to recommend the manuscript for publication in the present form.
Reviewer 2 Report
All comments are incorporated well